# Reverse Chromatin Immunoprecipitation (R-ChIP) enables investigation of the upstream regulators of plant genes

Xuejing Wen [1,2], Jingxin Wang[3], Daoyuan Zhang[1,2], Yu Ding[1,2,4], Xiaoyu Ji[5], Zilong Tan[1,2,4] & Yucheng Wang [1,2,3✉]

DNA binding proteins carry out important and diverse functions in the cell, including gene regulation, but identifying these proteins is technically challenging. In the present study, we developed a technique to capture DNA-associated proteins called reverse chromatin immunoprecipitation (R-ChIP). This technology uses a set of specific DNA probes labeled with biotin to isolate chromatin, and the DNA-associated proteins are then identified using mass spectrometry. Using R-ChIP, we identified 439 proteins that potentially bind to the promoter of the *Arabidopsis thaliana* gene *AtCAT3* (AT1G20620). According to functional annotation, we randomly selected 5 transcription factors from these candidates, including bZIP1664, TEM1, bHLH106, BTF3, and HAT1, to verify whether they in fact bind to the *AtCAT3* promoter. The binding of these 5 transcription factors was confirmed using chromatin immunoprecipitation quantitative real-time PCR and electrophoretic mobility shift assays. In addition, we improved the R-ChIP method using plants in which the DNA of interest had been transiently introduced, which does not require the T-DNA integration, and showed that this substantially improved the protein capture efficiency. These results together demonstrate that R-ChIP has a wide application to characterize chromatin composition and isolate upstream regulators of a specific gene.

[1] State Key Laboratory of Desert and Oasis Ecology, Xinjiang Institute of Ecology and Geography, Chinese Academy of Sciences, Urumqi 830011, China. [2] Turpan Eremophytes Botanical Garden, Chinese Academy of Sciences, Turpan 838008, China. [3] State Key Laboratory of Tree Genetics and Breeding, Northeast Forestry University, Harbin 150040, China. [4] University of Chinese Academy of Sciences, Beijing 100049, China. [5] College of Forestry, Shenyang Agricultural University, Shenyang 110866, China. ✉email: wangyucheng@ms.xjb.ac.cn

D NA–protein interactions comprise an important cellular process, playing fundamental roles in diverse biological functions, such as transcription regulation, transcription, and DNA replication, translation, repair, packaging, restriction and recombination[1,2]. Transcription factors (TFs) can activate or repress the expression of genes by interacting with genomic *cis*-regulatory DNA elements. Therefore, interactions between TFs and regulatory genomic DNA play a primary role in gene expression regulation, and are the backbone of gene regulatory networks[3]. Therefore, exploration of methods to investigate the interactions between DNA and proteins is an important aspect in molecular studies.

The methods to determine DNA–protein interactions can be classified into two complementary approaches, i.e., gene-centered (DNA-to-protein) approaches and TF-centered (protein-to-DNA) methods[4]. Gene-centered approaches are those in which one or more regulatory DNA element is used to identify the TFs that bind to these DNA sequences, such as yeast one-hybrid assays (Y1H). In contrast, the TF-centered approach uses a TF or set of TFs of interest to identify the genes that are regulated by these TF(s), and includes chromatin immunoprecipitation (ChIP), systematic evolution of ligands by exponential enrichment (SELEX), and DNA adenine methyltransferase identification (DamID)[5].

Transcriptional regulation is a critical process that is involved in almost all biological processes, including growth, development, metabolism, cell-cycle progression, cell division, and in response to biotic or abiotic stress[6,7]. In the gene expression process, TFs are the main effectors of transcription regulation. TFs regulate gene expression by binding to certain *cis*-acting elements present in the promoters of their regulated genes. Therefore, to reveal the regulatory mechanism of a gene's expression, it is important to identify its upstream regulators.

Usually, the gene-centered method is used to identify the upstream regulators of defined genes. Until now, limited gene-centered in vivo methods have been reported, including proteomics of isolated chromatin segments (PICh)[8], chromatin isolation by RNA purification (ChIRP)[9], protein–RNA interaction mapping assay (PRIMA)[10], and Y1H. Among these technologies, ChIRP and PRIMA are used to identify proteins that bind to RNA. Y1H is an excellent gene-centered approach; however, it cannot determine whether the interactions actually occur in cells. PICh can isolate specific chromatin segments together with all the associated proteins using a DNA as probe to hybridize with the specific chromatin, and mass spectrometry is performed to analyze the proteins[8]. However, this method is mainly used for Hela cells, but is rarely used in plants. In plants, Zeng et al.[11] isolated chromatin associated with the centromeres of barley using the PICh technique, and the proteins associated with centromeric chromatin were analyzed by mass spectrometry. D'Auria et al.[12] developed a technology to isolate and identify the proteins that bind to known DNA sequences using affinity capture combined with mass spectrometry. However, this method is used for microorganisms, and is also not suitable for plants. Therefore, an efficient in vivo gene-centered method with wide range of application should be developed for plants.

In the present study, we developed a method to capture proteins from specific formaldehyde cross-linked chromatin or DNA regions and to identify these proteins using mass spectrometric analysis (MS). Using this method, the proteins binding to the promoter region of the *Arabidopsis thaliana* gene *AtCAT3* (encoding Catalase 3) were isolated. We called this technology Reverse Chromatin Immunoprecipitation (R-ChIP), because it is the opposite of ChIP, using the DNA to retrieve the proteomic information.

## Results

**The procedure of R-ChIP.** The procedures of R-ChIP are shown in Fig. 1. In brief, the first step is crosslinking of proteins and chromatin using formaldehyde; the second step is isolation of nucleus; the third step is shearing the chromatin within 1 kb in length by sonication. In the fourth step, denaturation of chromatin with heat, and the fifth step is hybridization of the interested chromatin using specific probes. In the sixth step, harvest of the interested chromatin fragments using the affinity of biotin and streptavidin. The seventh step is wash and elution of the aim chromatin fragments, and the last step is analysis of the proteins associated with aim chromatin using mass spectrometer (Fig. 1). For building an efficient R-ChIP technology, some key steps, such as chromatin crosslinking, heat for denaturing chromatin and hybridization, are further studied in detail.

**Determination of the suitable hybridization buffer and temperature.** We first determined the suitable hybridization buffer for R-ChIP. The efficiencies of different hybridization buffers were determined, including high salt buffer (HS), medium salt buffer (MS), and low salt (LS) buffer, and the hybridization was conducted at 37 or 42 °C, with or without formamide. Probes labeled with biotin were used for DNA hybridization. After hybridization, the renatured DNA was collected using streptavidin beads, and the hybridization efficiencies were determined using qPCR. When using the HS hybridization buffer, the hybridization efficiency at 42 °C were higher than that at 37 °C with formamide (Fig. 2a). However, the hybridization efficiencies of HS are similar at 42 and 37 °C without formamide. When using MS buffer, the hybridization efficiencies were similar at 42 and 37 °C without formamide, but were substantially increased with formamide either at 37 or 42 °C (Fig. 2a). LS buffer displayed similar changing pattern of efficiency as MS did (Fig. 2a). The buffer supplied with 10% (v/v) formamide displayed higher hybridization efficiencies than those without formamide (Fig. 2a). Overall, the LS hybridization buffer showed highest hybridization efficiency, followed by the MS buffer, with the HS buffer showing lowest hybridization efficiency with formamide (Fig. 2a). In particular, the LS buffer showed the highest hybridization efficiency at 37 or 42 °C; therefore, LS supplemented with 10% (v/v) formamide was considered as the best buffer for DNA hybridization (Fig. 2a).

**Supplementation with sodium dodecyl sulfate (SDS) can increase the efficiency of DNA hybridization.** We studied whether SDS plays a role in hybridization. As HS hybridization buffer does not work well in hybridization (Fig. 2a), it was not considered for further study, and only MS and LS buffers were assessed. Different concentrations of SDS (0–1%, w/v) were supplied in LS or MS hybridization buffer. The results showed that increasing SDS from 0 to 1% (w/v) could gradually enhance the efficiency of DNA hybridization. However, the LS buffer displayed higher DNA hybridization efficiency compared with that of the MS buffer, whether supplied with SDS or not (Fig. 2b).

**The process of heat and hybridization will cause decross-linking of chromatin.** Decross-linking occurs during heating and hybridization; therefore, it is a relatively important factor that affects the capture of DNA associated proteins. We compared the effects of different hybridization buffers on chromatin crosslinking reversal during the heating and hybridization periods. Six hybridization buffers were studied, including MS and LS hybridization buffers supplied with 0, 0.5, and 1% (w/v) SDS. After denaturing for heat treatment and hybridization, the hybridization products were determined. The results showed that MS buffer induced chromatin

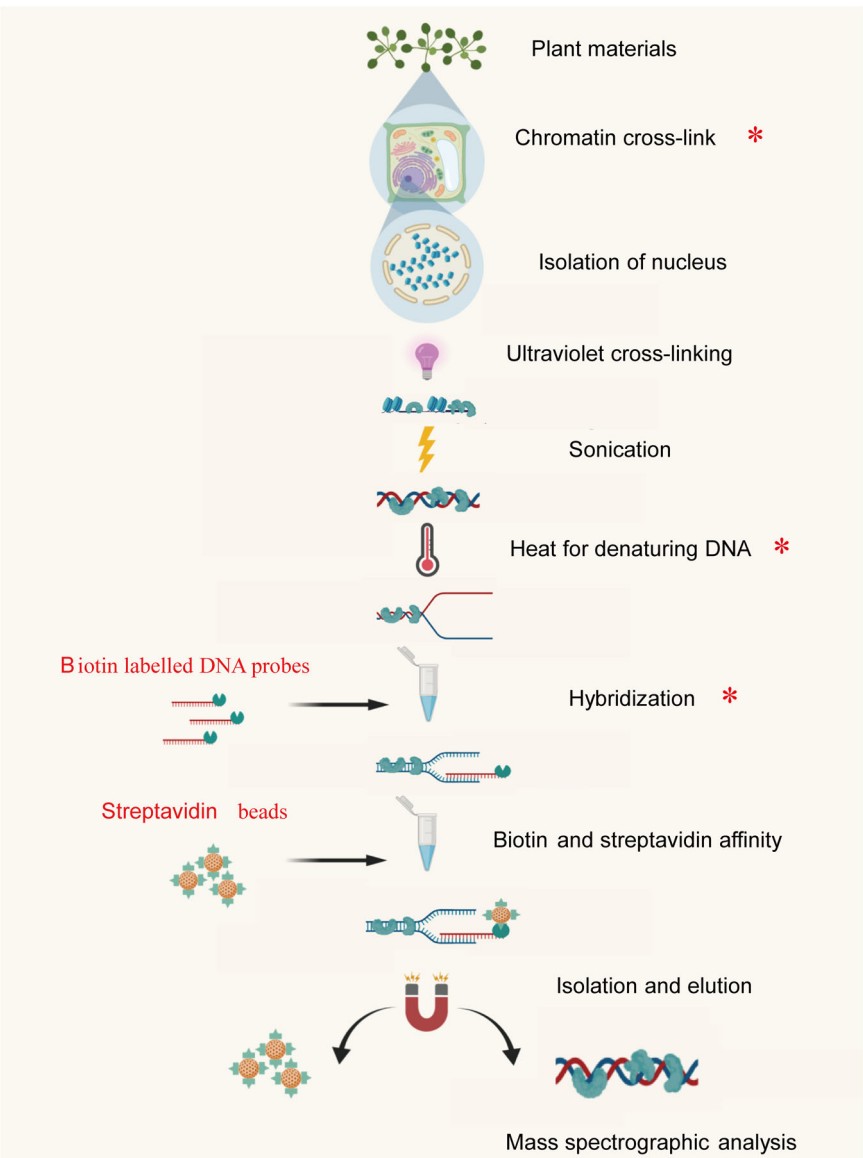

**Fig. 1 Outlines of the reverse ChIP procedure.** The plant material was treated with formaldehyde for DNA and protein cross-linking, and nuclei were isolated using sucrose density gradient centrifugation. Then the chromatin was sonicated to shear chromatin into small fragment. A set of probes labeled with biotin was added to the sonicated chromatin, and heated at 90 °C for 2 min to denature the chromatin so that the probes can bind to the denatured DNA sequence. Hybridization was then performed to anneal probes and chromatin. After hybridization, streptavidin magnetic beads were used to affinity the probes with biotin for collecting the hybridized chromatin. After washing to remove contamination, the hybridized chromatin was eluted, and DNA-associated proteins were subjected to mass spectrographic analysis. Asterisk (*) indicates the steps that need to be studied for optimization in this study.

decross-linking to a greater extent than LS hybridization buffer, whether supplemented with SDS or not (Fig. 3a). In addition, supplementation with SDS could increase chromatin decross-linking in both MS and LS buffers (Fig. 3a). In both LS and MS hybridization buffers, supplementation with 1% (w/v) SDS showed increased chromatin decross-linking compared with supplementation with 0.5% (w/v) SDS (Fig. 3a). Therefore, SDS induces decross-linking of chromatin during heating and hybridization. Although 1% (w/v) SDS showed the highest hybridization efficiency (Fig. 2b), it could also cause high level of protein and DNA decross-linking. Considering these factors together, 0.5% (w/v) SDS was chosen as suitable for use in hybridization.

**The effects of different concentrations of formaldehyde and $Ni^{2+}$ on preventing decross-linking of chromatin and proteins.** The effects of formaldehyde and $Ni^{2+}$ on preventing decross-

linking during heat treatment and hybridization were studied. The cross-linked DNA was treated at 90 °C for 2 min to denature chromatin, and hybridized for 4 h. The decross-linked chromatin was collected using Tris-phenol and chloroform extraction, and analyzed using qPCR. The results showed that 90 °C treatment and hybridization could lead to high levels of crosslink reversal. Compared with 1% (w/v) formaldehyde, 3% (w/v) formaldehyde treatment could substantially reduce decross-linking. In addition, 3% (w/v) formaldehyde combined with $Ni^{2+}$ could substantially reduce decross-linking of chromatin compared with 3% (w/v) formaldehyde crosslinking alone. Therefore, 3% (w/v) formaldehyde together with $Ni^{2+}$ was better to avoid decross-linking during the denaturation of chromatin in the R-ChIP procedure (Fig. 3b).

**The effects of heat and hybridization treatment on decross-linking.** Heat treatment is used to denature DNA, which is a

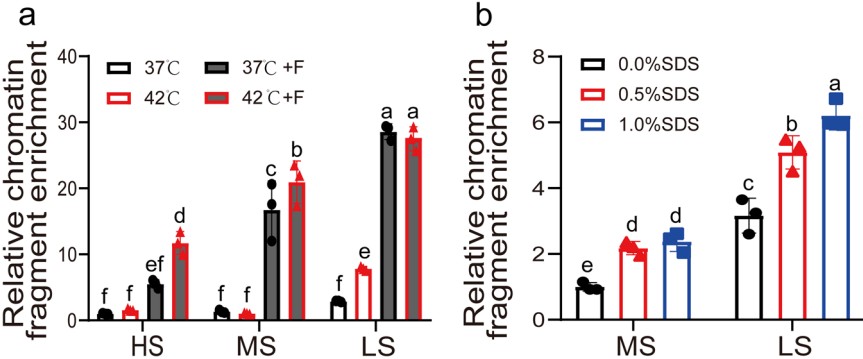

**Fig. 2 Determination of the suitable hybridization buffer for R-ChIP. a** Analysis of the efficiency of different hybridization buffers. Six hybridization buffers were studied, including HS (high salt), MS (medium salt), and LS (low salt) buffer, with or without 10% (v/v) formamide. Hybridization was performed at 37 °C or 42 °C. +F: supplement with 10% (v/v) formamide. The enrichment data using HS buffer at 37 °C was set as 1 to normalize other enrichment data including using the other hybridization buffers. **b** Analysis of the effects of SDS on DNA hybridization efficiency. MS and LS hybridization buffers containing 10% (v/v) formamide were, respectively, supplied with 0, 0.5%, or 1% (w/v) sodium dodecyl sulfate (SDS) for hybridization. The enrichment data using MS buffer and 0% (w/v) SDS was set as 1 to normalize the enrichment data using the other hybridization buffers. Three replicates (sample size of 50 seedlings) were performed ($n = 3$ independent experiments). Error bar indicates standard deviations of the mean measurements. Multiple comparisons of means were performed using Tukey's test, and different letters represent significant differences among treatments ($P < 0.05$). The promoter region of *AtCAT3* used for the qPCR evaluation was site 1 and is shown in Fig. 7l and Supplementary Data 4. The promoter of *AtActin3* was used as the internal control (which is far away from the promoter of *AtCAT3*).

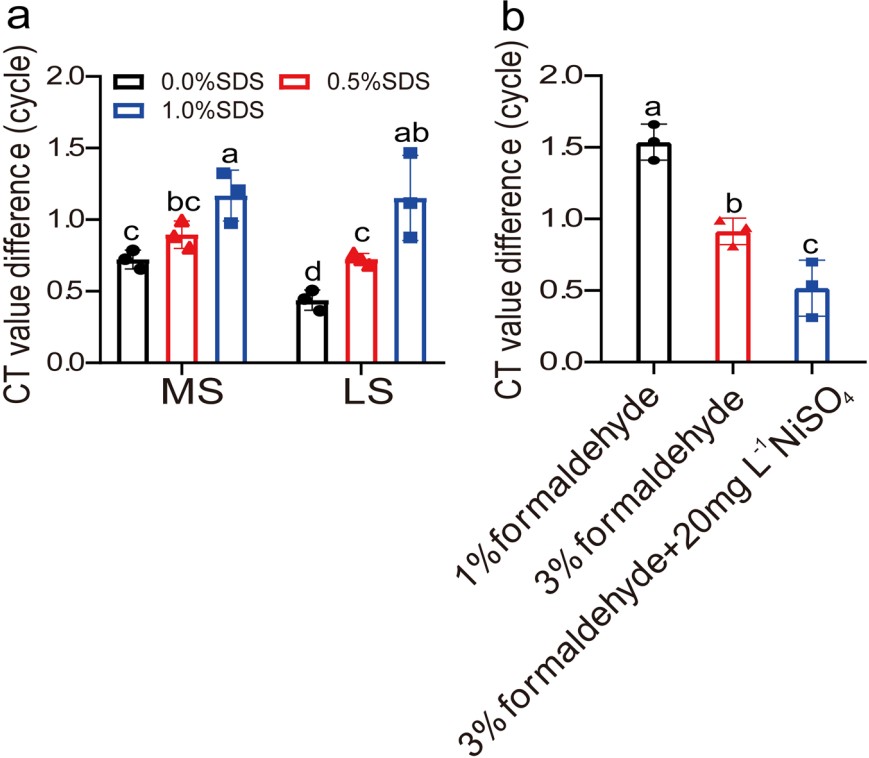

**Fig. 3 Analysis of the effects of different hybridization buffers and cross-linking methods on decross-linking of chromatin. a** Analysis of the effects of SDS concentration and different buffers on decross-linking of chromatin. MS and LS hybridization buffer [containing 10% (v/v) formamide supplied with 0, 0.5%, and 1% (w/v) SDS] were used in hybridization at 37 °C for 4 h. To analyze decross-linking, the chromatin was cross-linked with 3% (w/v) formaldehyde, and treated at 90 °C for 2 min for denaturation before hybridization. **b** Analysis of the effects of different crosslinking methods on preventing decross-linking of chromatin during heat and hybridization. The chromatin was cross-linked with 1%, 3%, or 3% (w/v) formaldehyde supplement with 20 mg L$^{-1}$ NiSO$_4$, and incubated in LS buffer [with 10% (v/v) formamide, 0.5% (w/v) SDS] at 90 °C for 2 min to denature the chromatin, and then hybridized for 4 h. After heating and hybridization, the decross-linked DNA was quantified using qPCR. The y-axis indicates the CT value difference (i.e., ΔCt), ΔCt = Ct (decross-linked DNA before heat) − Ct (decross-linked DNA after hybridization). Increased ΔCt value indicates increased decross-linking of DNA and protein. Three replicates (sample size of 50 seedlings) were performed ($n = 3$ independent experiments). Error bar indicates standard deviations of the mean measurements. Multiple comparisons of means were performed using Tukey's test, and different letters represent significant differences among treatments ($P < 0.05$). The promoter region of *AtCAT3* used for the qPCR evaluation was site 1 and is shown in Fig. 7l and Supplementary Data 4.

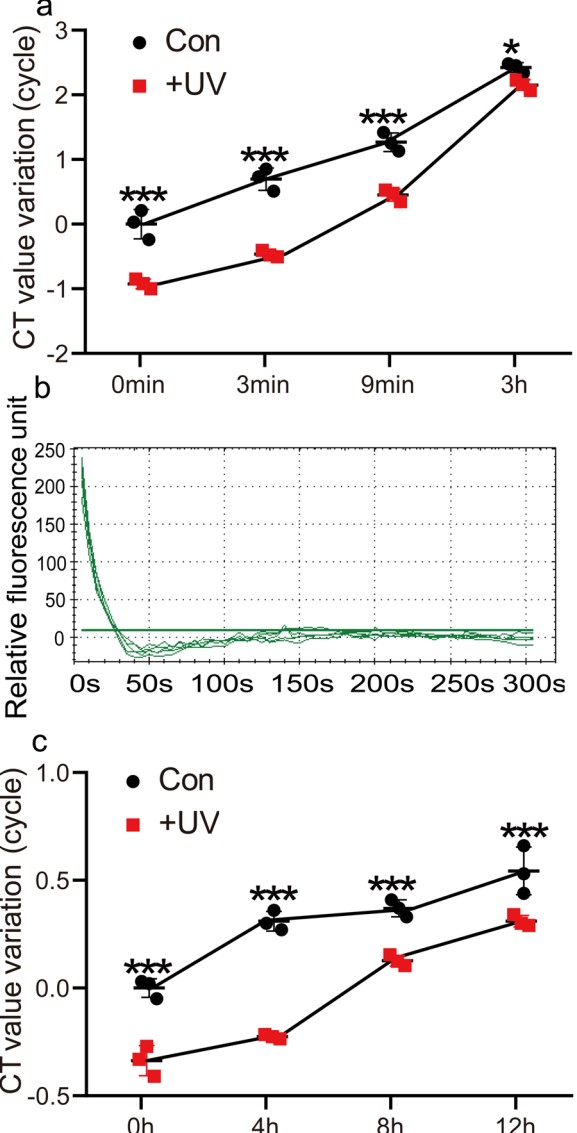

**Fig. 4 The effects of heat treatment and hybridization on denaturing cross-linked chromatin and decross-linking. a** Analysis of the decross-linking of chromatin after exposure to heat treatment. Two methods for crosslinking chromatin were used, including 3% (w/v) formaldehyde and 20 mg L$^{-1}$ NiSO$_4$ (Con), and 3% (w/v) formaldehyde and 20 mg L$^{-1}$ NiSO$_4$ combined with ultraviolet radiation treatment (UV). The cross-linked chromatin was incubated in LS buffer [with 10% (v/v) formamide, 0.5% (w/v) SDS] at 90 °C for different times, and the decross-linked DNA was quantified using qPCR. Increased Ct value variation indicates increased decross-linking of DNA and protein. **b** Determination of the time required for denaturing the cross-linked chromatin. After sonication, the sonicated cross-linked chromatin was added with SYBR Green and incubated at 90 °C. The denatured chromatin was monitored at 5 s intervals by visualization with SYBR Green fluorescence using a real-time PCR instrument. Five replicates were performed ($n = 5$ independent experiments). The reduction of fluorescence reflected the denaturation of chromatin. **c** Analysis of the decross-linking of chromatin during hybridization. The chromatin cross-linked with 3% (w/v) formaldehyde combined with NiSO$_4$ (Con), and the same cross-linked chromatin further cross-linked with ultraviolet radiation (UV) before sonication were used for analysis. These two kinds of cross-linked chromatin were incubated in LS buffer [with 10% (v/v) formamide, 0.5% (w/v) SDS] at 37 °C for 12 h, and the decross-linked DNA in different time points were harvested by Tris-phenol and chloroform extraction, and quantified using qPCR. The y-axis indicates the CT value variation (i.e., ΔCt), ΔCT = Ct decross-linked DNA in the control at 0 h [cross-linked with 3% (w/v) formaldehyde together with 20 mg L$^{-1}$ NiSO$_4$]− Ct decross-linked DNA of samples [including chromatins crosslinked with 3% (w/v) formaldehyde combined with NiSO$_4$ and 3% (w/v) formaldehyde combined with NiSO$_4$ and ultraviolet radiation] at different time points. Therefore, ΔCT > 0 indicates increased decross-linking of chromatin. ΔCT < 0 means decreased decross-linking of chromatin. The amount decross-linked chromatin in the sample cross-linked with formaldehyde together with ultraviolet radiation was less than that in the control; therefore, its ΔCt value was negative. Three replicates (sample size of 50 seedlings) were performed ($n = 3$ independent experiments). Error bar indicates standard deviations of the mean measurements. Asterisks indicate significant difference between Con and UV treatment (*$P < 0.05$, ***$P < 0.001$, t-test). The promoter region of AtCAT3 used for the qPCR evaluation was site 1 and is shown in Fig. 7l and Supplementary Data 4.

necessary step for DNA hybridization. LS hybridization buffer was found to be the best hybridization buffer in R-ChIP (Fig. 2a); therefore, we further studied whether 90 °C treatment could lead to decross-linking of chromatin in LS hybridization buffer [supplied with 10% (v/v) formamide and 0.5% (w/v) SDS]. After 90 °C treatment, the decross-linked DNA was recovered using Tris-phenol and chloroform extraction, and analyzed using qPCR. The results showed that 90 °C treatment could lead to decross-linking of DNA and proteins, and decross-linking was increased with the duration of heat treatment (Fig. 4a). We further studied how long it takes to denature the cross-linked chromatin completely. The degree of denaturation of cross-linked chromatin was determined by the binding of SYBR Green to DNA, and visualized using real-time PCR. The results showed that 90 °C treatment for 40 s could completely denature the cross-linked chromatin (Fig. 4b). Taken together, these results showed that, to guarantee complete denaturation of cross-linked chromatin and avoid high levels of decross-linking, we chose the following chromatin denaturation conditions: 90 °C for 1–2 min, because 90 °C treatment for less than 2 min does not cause high levels of chromatin decross-linking (Fig. 4a).

We next studied whether DNA–protein decross-linking occurred during hybridization. The chromatin was cross-linked

with 3% (w/v) formaldehyde and 20 mg L$^{-1}$ NiSO$_4$ was used as a control. Hybridization could lead to substantially decross-linking of DNA and proteins, and the amount of decross-linked chromatin was observed to increase with increasing of hybridization time (Fig. 4c).

**Heat and hybridization induce decross-linking that can be alleviated using ultraviolet radiation crosslinking.** In consideration of the above results that heat treatment and hybridization could cause severe decross-linking, it was necessary to reduce decross-linking of the cross-linked chromatin during heat and hybridization. Before sonication, the cross-linked chromatin was further cross-linked using ultraviolet radiation, and then was used for heat treatment and hybridization. The results showed that further cross-linking using ultraviolet radiation could markedly reduce crosslinking reversal during heat treatment (Fig. 4a) and hybridization (Fig. 4c) compared with no ultraviolet treatment. Therefore, to reduce decross-linking during heat and hybridization, ultraviolet radiation treatment should be performed before sonication that will increase the efficiency of protein capture (Fig. 4a, c).

**The key factors for building efficient reverse ChIP.** The results presented so far showed that: (1) LS buffer supplied with 10%

(v/v) formamide and 0.5% (w/v) SDS is the best hybridization buffer; (2) for crosslinking chromatin, 3% (w/v) formaldehyde together with $20\,mg\,L^{-1}\,Ni^{2+}$ should be used, and ultraviolet crosslinking should be further performed before chromatin sonication; (3) to denature chromatin DNA and avoid decross-linking during chromatin denaturation, the cross-linked chromatin should be incubated in LS buffer containing 10% (v/v) formamide and 0.5% (w/v) SDS, and heated at 90 °C for 2 min.

The more detailed steps were build based on the main procedures of R-ChIP (Fig. 1). In brief, plants were cross-linked with 3% (w/v) formaldehyde and $20\,mg\,L^{-1}\,NiSO_4$, and their nuclei were purified. To reduce decross-linking, ultraviolet cross-linking of chromatin was performed before sonication. The chromatin was then sheared into DNA fragments within 1 kb using sonication. After sonication, a set of single strain DNA probes were added, and the chromatin was denatured by heat treatment at 90 °C for 2 min. Hybridization was then performed in LS buffer supplied with 10% (v/v) formamide and 0.5% (w/v) SDS at 37 °C. After hybridization, streptavidin magnetic beads were added to collect the hybridization products by biotin and streptavidin affinity. The magnetic beads were the washed and the hybridization products were eluted. The hybridization efficiency was analyzed using qPCR. If the efficiency of hybridization was good according to the result of qPCR, the captured proteins were further analyzed using mass spectrometry.

**Determination of the efficiency of transient transformation.** In order to further increase the efficiency of R-ChIP, transient transformation was performed to increase the abundance of target promoter in plant cells. The efficiency of transient transformation was first determined by GUS staining of the Arabidopsis plants that were transiently transformed with empty pCAMBIA1301. The results showed that GUS activity could be detected throughout the Arabidopsis plants, indicating that transient transformation had been successful (Fig. 5a; the full and uncropped image is shown in Supplementary Fig. 1). To further determine the efficiency of transient transformation in the materials used for R-ChIP, the plants were transiently transformed with pBI121-ProCAT:AtCAT3-GUS for qRT-PCR analysis. The results showed that the transcripts of AtCAT3 were highly elevated in transiently transformed plants compared with those in the wild-type plants, suggesting that the transient

transformation had been successfully performed, and could be used in further study (Fig. 5b).

As transient transformation could substantially increase the expression of gene (Fig. 5b), suggesting that the introduced T-DNA had been bound with the regulatory proteins, which makes it is possible to capture the regulatory proteins using the transiently transformed plants as the start material.

**Determination of the efficiency of R-ChIP.** To capture the upstream regulators of AtCAT3, R-ChIP was performed with two kinds of materials, i.e., transiently transformed plants harboring T-DNA that contained the promoter of AtCAT3 (transiently transformed) and wild-type plants. A set of probes with biotin were used to capture the region of promoter of AtCAT3, and the length and location information of these probes are provided in Supplementary Fig. 2.

The efficiency of capturing DNA binding proteins was determined by calculating the relative abundance of the studied DNA fragments using qPCR. The relative abundance in wild-type plants was 371-fold (Fig. 6a). However, in the transiently transformed plants, the relative abundance was 823-fold (Fig. 6a). The relative abundance of the studied DNA from the transiently transformed plants was 2.2 times higher of that harvested from wild-type plants. This result indicated that the transiently transformed plants were more efficient in R-ChIP isolation.

To test the repeatability of R-ChIP, three biological replications were performed for the transiently transformed and wild-type plants. The captured proteins in each biological replication were visualized using SDS-PAGE stained by silver nitrate (Fig. 6b; the full and uncropped gel image is shown in Supplementary Fig. 3).

**R-ChIP analysis of the proteins that bind to the AtCAT3 promoter.** Mass spectrometry analysis was performed to detect the proteins isolated using R-ChIP, and the results of 6 samples are provided in Supplementary Data 1. The biological replicate results showed that 1707, 1811, and 1872 putative unique proteins in total were identified from three replications of wild-type Arabidopsis material; meanwhile, 1392, 1706, and 1571 putative unique proteins were identified from three replicates of transiently transformed material (Table 1).

The nuclear proteins were analyzed from among the total identified proteins. There were 439, 460 and 462 putative unique

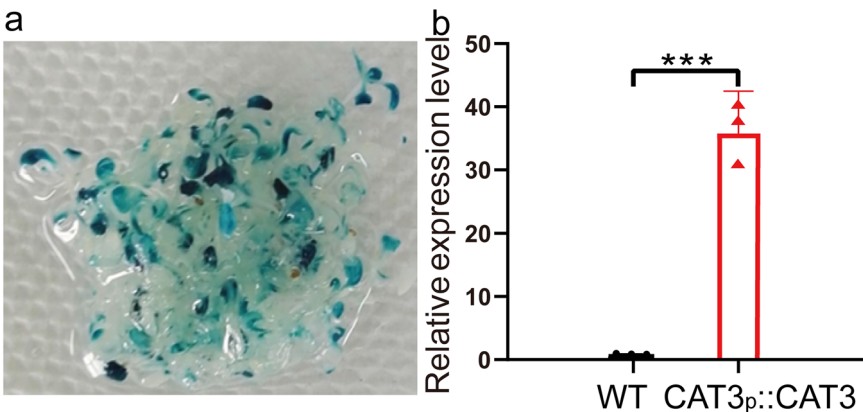

**Fig. 5 Determination of the efficiency of transient transformation. a** Investigation of the efficiency of transient transformation by GUS staining. The Arabidopsis plants were transiently transformed with pCAMBIA1301. After transformation for 72 h, GUS staining was performed. **b** Determination of the expression of AtCAT3 gene in the plants transformed with pBI121-ProCAT3:CAT3-GUS. WT: wild-type plants:CAT3p::CAT3:transient transformed plants. The primers for amplification of AtCAT3 were used in quantitative real-time reverse transcription PCR, and the AtActin3 was used as the internal control. Three replicates (sample size of 50 seedlings) were performed (n = 3 independent experiments). Error bar indicates standard deviations of the mean measurements. Asterisks indicate highly significant difference between two samples (***P < 0.001, t-test).

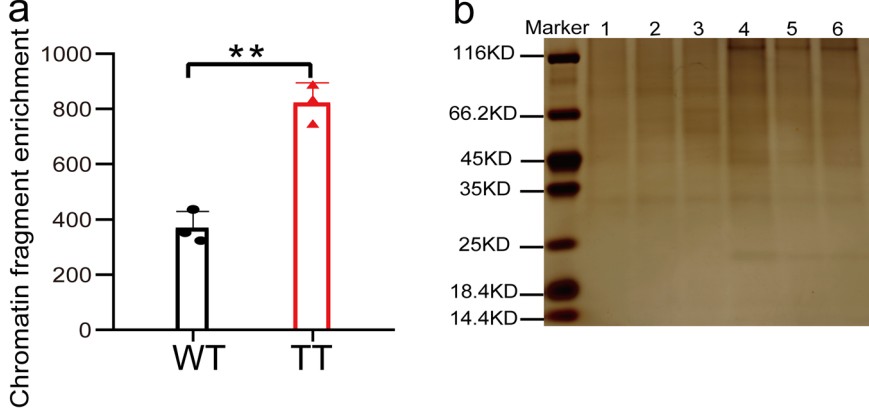

**Fig. 6 Analysis of the captured proteins and the relative enrichment of DNA using R-ChIP. a** Relative enrichment analysis by R-ChIP from wild-type or transiently transformed plants. WT: wild-type plants; TT: transiently transformed plants.The relative enrichment was calculated as $2^{-(\Delta CT 1 - \Delta CT 2)}$. $\Delta$CT 1: The Ct of the studied (isolated) DNA subtracted from the Ct of the internal control DNA in the products of R-ChIP. $\Delta$CT 2: The Ct of DNA of studied (isolated) subtracted from the Ct of the internal control DNA in the input sample. The promoter region of *AtCAT3* used for the qPCR evaluation was site 1 and is shown in in Fig. 7l and Supplementary Data 4. The promoter of *AtActin3* (which is far away from the promoter of *AtCAT3*) was used as the internal control. Three replicates (sample size of 500 seedlings) were performed (*n* = 3 experiments). Error bar indicates standard deviations of the mean measurements. Asterisks indicate highly significant difference between two samples (**$P$ < 0.01, *t*-test). **b** SDS-PAGE analysis of the proteins captured using R-ChIP. Lanes 1–3: the proteins isolated from wild-type plants from three biological replicates, respectively. Lanes 4–6: the proteins isolated from transient transformed plants from three biological replicates, respectively. The proteins were stained with silver nitrate.

**Table. 1 Identification of the DNA associated proteins using R-ChIP.**

|  | Wild-type samples | | | Transient transformed samples | | |
| --- | --- | --- | --- | --- | --- | --- |
|  | Sample 1 | Sample 2 | Sample 3 | Sample 4 | Sample 5 | Sample 6 |
| Total proteins | 1707 | 1811 | 1872 | 1392 | 1706 | 1571 |
| Nuclear proteins | 439 | 460 | 462 | 334 | 411 | 384 |
| Transcription regulators | 89 | 92 | 90 | 94 | 107 | 113 |
| Common proteins | 1061 | 1008 | 1058 | 630 | 638 | 628 |
| Common nuclear proteins | 138 | 130 | 128 | 69 | 68 | 67 |
| Common transcription regulators | 14 | 17 | 16 | 14 | 15 | 14 |

Sample 1–3: biological replicates of wild-type plants; sample 4–6: biological replicates of transiently transformed plants.

nuclear proteins identified from wild-type samples, which accounted for 24.7–25.7% of total identified proteins (Table 1). At the same time, 334, 411, and 384 putative unique nuclear proteins were identified from the transiently transformed samples, which accounted for 24.0–24.4% of total identified proteins (Table 1). Among the nuclear proteins, 89, 92, and 90 TFs were identified from the wild-type samples, 94, 107, and 113 TFs were detected in the transiently transformed samples (Table 1). The TFs identified from wild-type plants accounted for 4.8–5.2% of total identified proteins; however, the TFs identified from transiently transformed plants accounted for 6.3–7.2% of total identified proteins. These results indicated that use of transiently transformed plants as the research material in R-ChIP could yield more TFs.

**Confirmation of R-ChIP results using ChIP analysis**. We obtained 439 transcription regulators from wild-type and transiently transformed samples in total, including several principal transcription factor families that play important roles in stress tolerance. For example, myeloblastosis oncogene transcription factor (MYB) family (17 unique members identified), WRKY transcription factor (WRKY) family (14 unique members identified), basic helix-loop-helix transcription factor (bHLH) family (8 unique members identified), NAM/ATAF/CUC transcription factor (NAC) family (8 unique members identified), and basic leucine zipper

transcription factor (bZIP) family (6 unique members identified). The list of transcription regulators obtained by mass spectrometry is provided in Supplementary Data 2. To further determine whether the identified TFs really bind to the promoter of *AtCAT3*, five identified TFs were randomly selected for further study. Among them, two were only identified from transiently transformed samples, and three were identified from both wild-type and transiently transformed plants, including bZIP1664 (AT1G24267), APE-TALA2/ ERE binding factor:TEMPRANILLO1 (TEM1, AT1G25560), bHLH106 (AT2G41130), Basic transcription factor 3 (BTF3, AT1G17880), and homeodomain-leucine zipper (HD-ZIP) transcription factor (HAT1, AT4G17460). *Arabidopsis* plants transiently transformed with 35S:TF-FLAG were generated for the ChIP analysis. First, the expression of these transgenes in the transiently transformed plants were studied. Compared with the control plants, these transgenes all displayed substantially increased expression levels (Fig. 7a), suggesting that these genes had been successfully transformed, and could be used in ChIP study. ChIP-qPCR was conducted to determine whether the bindings of bZIP1664, TEM1, bHLH106, BTF3, and HAT1 to the *AtCAT3* promoter actually occurred in *Arabidopsis* plants. The results showed that the truncated promoter of *AtCAT3* was substantially enriched for all five TFs, suggesting that the *AtCAT3* promoter was bound by bZIP1664, TEM1, bHLH106, BTF3, and HAT1 in *Arabidopsis* plants (Fig. 7b–f). The enrichment folds of all studied TFs in *AtCAT3* promoter are shown in Supplementary Data 3.

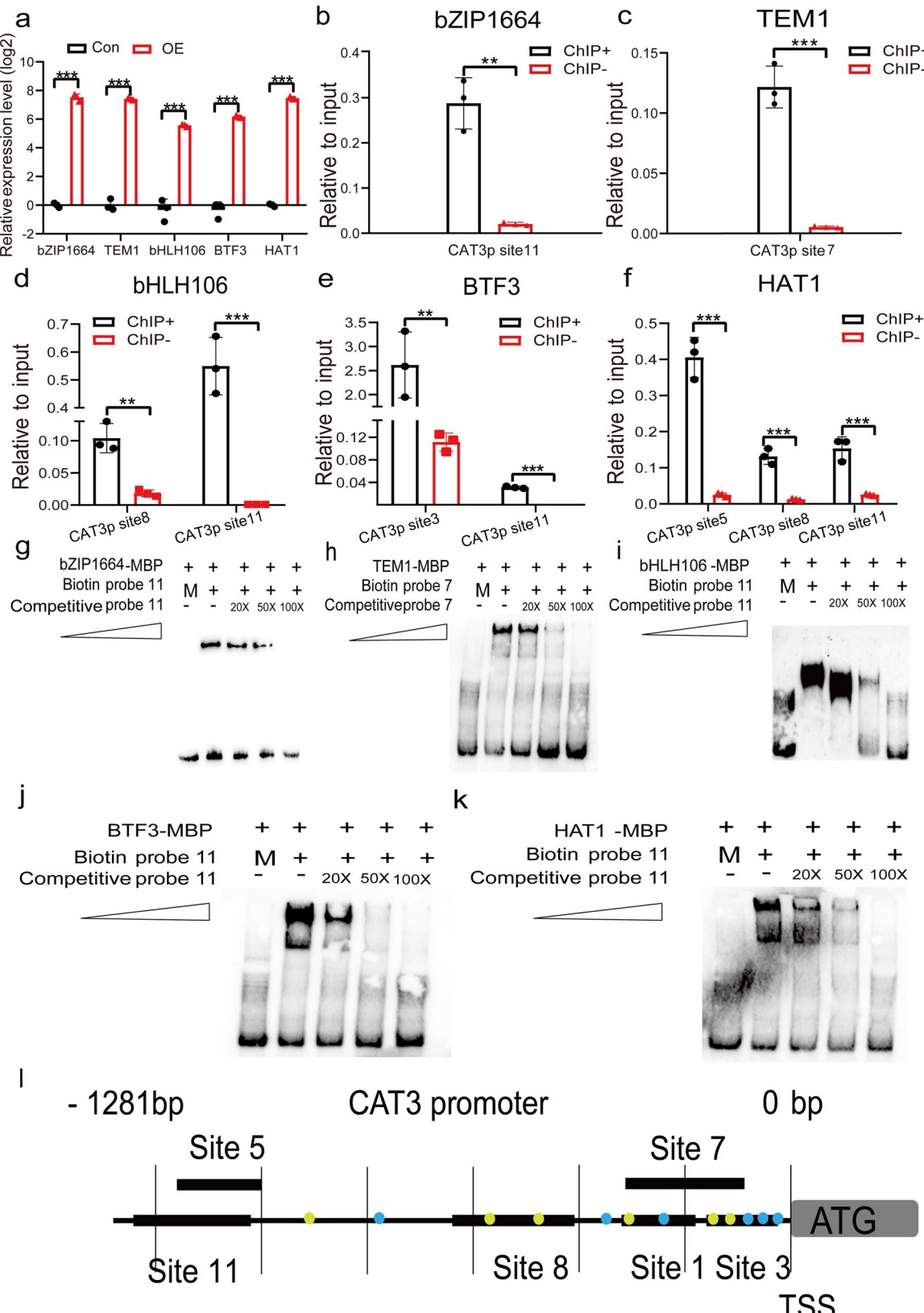

**Electrophoretic mobility shift assay (EMSA).** To further confirm the binds of these five TFs to the promoter of *AtCAT3*, EMSA was performed. The different truncated promoters of *AtCAT3* that were identified to be bound by the five studied TFs in ChIP-qPCR were used as probes in EMSA, and the information of their length and location are shown as Supplementary Data 4. The interaction of bZIP1664, TEM1, bHLH106, BTF3, and HAT1 proteins to their respective DNA probes were studied by EMSA. The results showed that these five TFs could all form DNA–protein complex bands when interacting with the truncated promoter of *AtCAT3*. In addition, when the unlabeled probe competitor was added, the signal intensities of DNA–protein complex bands were gradually

**Fig. 7 Verification of the identified upstream regulators of *AtCAT3*. a** The expression of transgenes in the transiently transformed *Arabidopsis* plants. **b–f** The binding of TFs to the promoters of *AtCAT3* as assessed using ChIP-qPCR. Only significant enrichments are shown in (**b–f**), and the enrichment of all these TFs in different regions of *AtCAT3* promoter are in Supplementary Data 3. ChIP+: an anti-Flag antibody was used to immunoprecipitate the target chromatin; ChIP−: chromatin without antibody immunoprecipitation served as a negative control. **g–k** The binding of TFs to the promoters of *AtCAT3* as assessed using EMSA. M: mutant probe was the promoter of *AtActin3*. **l** The diagram of the promoter region of *AtCAT3*. Yellow dots indicate the forward probes. Blue dots indicate the reverse probe. Sites 1, 3, 5, 7, 8, and 11 indicate different regions of the promoter of *AtCAT3*, respectively. Three replicates (sample size of 100 seedlings) were performed ($n = 3$ independent experiments). Error bar indicates standard deviations of the mean measurements. Asterisks indicate highly significant difference between OE and Con (**b**) or ChIP+ and ChIP− (**c**) (**$P < 0.01$, ***$P < 0.001$, *t*-test).

decreased (Fig. 7g–k; full and uncropped blot images are in Supplementary Figs. 4–8), and the mutated versions of the probes could not form the DNA–protein complex bands. These results indicated that these five TFs actually bind to the *AtCAT3* promoter, further verifying the reliability of R-ChIP results.

## Discussion

The detection of proteins that interact with chromatin regions or DNA sequences of interest is important to increase our knowledge of genome biology. Although these questions have been studied widely, we still lack a powerful method to provide information about the complete composition of a locus, especially for plants. In the present study, we developed a method to capture proteins, which was termed R-ChIP. This method has the potential to examine composition changes and all interacting proteins during regulation. In addition, combined with the transient genetic transformation method, this method could efficiently identify the upstream regulators of a gene of interest.

We used the developed R-ChIP method to identify the proteins that bind to the promoter of *AtCAT3*. The *AtCAT3* gene was used because it is well characterized, and encodes a protein with CAT activity that plays a role in drought tolerance[13]. Therefore, revealing its upstream regulators might be helpful to further characterize its mechanism in drought tolerance.

In the present study, we found that decross-linking of chromatin is occurred during the process of R-ChIP, including sonication, denaturing chromatin and hybridization, which was closely involved in the success of R-ChIP (Figs. 3, 4). The suitable hybridization buffer was first determined to reduce decross-linking of chromatin (Fig. 3a); then the crosslinking methods were selected to reduce decross-linking of chromatin during heat treatment and hybridization (Fig. 3b). As heat treatment can cause high decross-linking of chromatin (Fig. 4a), the suitable time for denaturing DNA were determined (Fig. 4b). In addition, the method to further prevent decross-linking of chromatin was studied during heat and hybridization (Fig. 4a, c). All these studies built a robust method to reduce decross-linking of chromatin, which will highly increase the success of R-ChIP.

Previously, a method termed proteomics of isolated chromatin segments (PICh) was developed, which could isolate genomic DNA with its associated proteins using a specific nucleic acid probe[8]. Both PICh and R-ChIP are gene-centered methods to capture proteins bound to a known DNA sequence, and the strategy of R-ChIP is similar to that of PICh. However, these two methods are quite different in many aspects. The main differences are as the follows. (1) The materials used for PICh are Hela cells, but the materials used in R-ChIP are plants. (2) R-ChIP includes the isolation and purification of cell nuclei to enrich the nuclear proteins; however, PICh does not have this step. (3) The hybridization buffer and hybridization steps were quite different between the two methods. The hybridization buffer of PICh is LBJD solution (10 mM HEPES-NaOH, pH 7.9; 100 mM NaCl; 2 mM EDTA, pH 8; 1 mM EGTA, pH 8; 0.2% SDS; 0.1% Sarkosyl, protease inhibitors), which is quite different from that of R-ChIP. (4) The washing procedures are also different. (5) Previous

reports showed that PICh is not suitable for DNA sequences present as a single copy or at a low copy number in the genome and needs a large amount starting material[8,12].

Nuclear proteins account for a low percentage of total proteins. For instance, in *Arabidopsis*, nuclear proteins account for approximately 20% of all genome-encoded proteins and 15% of experimentally identified proteins; therefore, it is difficult to detect nuclear proteins accurately because of their low abundance in the global proteome[14,15]. In addition, TF proteins only account for a low percentage of nuclear proteins, increasing the difficulty of identifying them using mass spectrometry. In the present study, we found that nuclear proteins accounted for 24.2% of total proteins identified from the transiently transformed samples, and accounted for an average of 25.3% of total proteins identified from the wild-type materials (Table 1). In addition, the identified TFs, respectively, accounted for 5.0% and 6.7% of total proteins identified from wild-type and transiently transformed plants (Table 1). These results indicated that R-ChIP is efficient in capturing DNA associated proteins, and using transient transformed plants to capture TFs is more efficient than using wild-type plants. These results together suggested that R-ChIP can be used in detection of the upstream regulators of an interested gene, especially when using transiently transformed plants.

Our studies showed that the repetition rates of three biological repeats were from 55.7% to 62.2% in wild-type samples and 37.4% to 45.3% in transiently transformed samples. The low repetition rates of transiently transformed samples may be due to the reason that the efficiencies of transient transformation are varied in different biological repeats. In wild-type samples, repetition rates were about 60%, and not very high. This should be due to the reason that when the abundance of protein is lower to a certain degree, it may be detected by one mass spectrometer analysis, but may not be detected by another analysis because of the sensitivity of mass spectrometer. In this study, the low abundance of nuclear proteins, especially the low abundance of TFs, which leads to the low repetition rates between two biological repeats. However, the aim of R-ChIP is to detect the presence of proteins. Due to the low abundance of TFs and the sensitivity of MS spectrometer, detection by one biological repeat is enough to indicate their presence. Our results also showed that the TFs detected by only one biological repeat are also really the upstream regulators (Fig. 7). Therefore, the three biological repeats are necessary in R-ChIP, but it should be used for detection of more TFs rather than to study the repeatability of experiment.

Our results showed that transient transformation could increase the expression of gene (Fig. 5a); however, T-DNA should not be integrated into genome during this period. This phenomenon indicated that the bindings of regulatory proteins to the *cis*-acting elements contained in T-DNA does not require T-DNA integration. Therefore, the plants with transient transformation could be used in protein capture without considering whether T-DNA had been integrated into genome. In the present study, the results showed that R-ChIP could enrich the DNA fragment of interest by nearly 371-fold using wild-type plants as the material,

but enriched it by 823-fold using the transiently transformed plants (Fig. 6a). These results indicated that R-ChIP could highly enrich the DNA sequence of interest, indicating that this technology had been developed successfully. In particular, the enrichment by R-ChIP increased when using the transiently genetically transforming plants. The percentage of TFs among the total proteins isolated from wild-type plants was 5.0%; while percentage of TFs from transiently transformed plants increased to 6.8%, indicating that using transiently transformed plants can highly enrich TFs (Table 1).

Furthermore, to verify the reliability of R-ChIP, TFs were selected from the transiently transformed and wild-type plant data. ChIP and EMSA were then performed, which indicated that all the randomly selected TFs were genuine upstream regulators of the AtCAT3 promoter. The use of the transiently transformed plants equally increases the copy number of studied DNA as material, which will improve the efficiency of R-ChIP. In addition, theoretically, the transient transformation can be used in any plant species that can be transformed by Agrobacterium tumefaciens, and is a simple and fast technique[16,17]. Taken together, the results suggested that R-ChIP are performed using transiently transformed plants is a powerful method to detect of upstream regulators of a gene of interest. One drawback of R-ChIP combined with transient transformation is that it might not be suitable for chromatin locus-specific composition, because the transferred T-DNA cannot exactly reflect the chromatin status. However, this problem can be solved using wild-type plants and increasing the amount of starting material.

## Methods

**Plant material.** The *A. thaliana* ecotype Columbia background was used in this study. The sterilized seeds were plated on Murashige–Skoog medium [supplemented with 2% (w/v) sucrose and 0.6% (w/v) agar] in the tissue culture room with the temperature of 23 °C and photoperiod of 16 h/8 h (day/night). After growth for 3 weeks, these plants were harvested and used as the study material.

**Construction of the expression vector and the method of transient transformation.** The promoter sequence (1281 bp upstream from the translation start site) of *AtCAT3* (At1G20620) was cloned into vector pBI121 to replace the 35S promoter (pBI121-ProCAT), and the coding region (CDS) of *AtCAT3* was further cloned into pBI121-ProAtCAT3 to fuse with its *GUS* gene (pBI121-ProCAT: AtCAT3-GUS) (Supplementary Data 6). This construct was used for transient transformation to investigate the upstream regulators of *AtCAT3*. Transient transformation was performed according to Zang et al.[18] with minor modifications. In brief, the colonies of *A. tumefaciens* EHA105 harboring pBI121-ProAtCAT3, or empty pCAMBIA1301, were cultured to an optical density at 600 nm (OD600) of 0.6–0.8 with shaking at 180 rpm at 28 °C. The cultures were centrifuged at 3000 × *g*, and adjusted to an OD600 of 1.2 using transformation solution [5% (w/v) sucrose, 150 μM acetosyringone, 5 mM CaCl$_2$, 0.015% (w/v) DTT, 20 μM 5-AZA, 0.02% (v/v) Tween20]. Whole *Arabidopsis* seedlings (3 weeks old) were soaked in 100 mL of transformation solution with shaking at 100 rpm for 3 h at 25 °C and then wiped with sterile filter papers to remove excess solution. Then seedlings were cultured on Murashige–Skoog medium [2% (w/v) sucrose, pH 5.8] for 72 h. To determine the transient transformation efficiency, GUS staining was performed on the seedlings transient transformed with empty pCAMBIA1301, following the method of Jefferson et al.[19].

**Design of the probes to isolate the *AtCAT3* promoter.** In total, 12 probes were designed according to the promoter sequence of *AtCAT3* (1281 bp), each DNA strand had six matched probes, and their distribution is shown in Fig. 7a. All the probes (20 nt) were directly synthesized with biotin labeling at their 5′ terminus. The sequences and locations of these probes are shown in Supplementary Data 7 and Supplementary Fig. 2.

**Determination of the suitable hybridization buffer and temperature.** Different hybridization buffers were used, including HS buffer [30 mM Tris-HCl (pH 7.4), 500 mM NaCl], MS buffer [30 mM Tris-HCl (pH 7.4), 150 mM NaCl], and LS buffer [30 mM Tris-HCl (pH 7.4), 20 mM KCl, 10 mM(NH$_4$)$_2$SO$_4$, 3 mM MgCl$_2$]. All these buffers contained 1 mM phenylmethylsulfonyl fluoride (PMSF) and 1 μg mL$^{-1}$ proteinase inhibitors, 0.5% (w/v) SDS, and were supplied with 10% (v/v) formamide or not. The DNA probes with biotin were added to a final concentration of 0.6 μM each, and incubated at 37 or 42 °C for 4 h. Then, the isolation

of hybridized DNA, washing and elution of the beads were performed according to step 12–16 of the R-ChIP protocol.

Analysis of the hybridization efficiency was performed according to step 18 of the R-ChIP protocol. Real-time PCR was performed using specific primers to detect the hybridization efficiency (Supplementary Data 8).

**The effects of SDS on DNA hybridization efficiency.** To determine the effects of SDS on the efficiency of DNA hybridization, the hybridization buffers were supplied with different concentrations of SDS, including MS buffer [supplied with 10% (v/v) formamide; 1 mM PMSF; proteinase inhibitors 1 μg mL$^{-1}$; and 0, 0.5, or 1% (w/v) SDS] and LS buffer [supplied with 10% (v/v) formamide; 1 mM PMSF; proteinase inhibitors 1 μg mL$^{-1}$; and 0, 0.5, or 1% (w/v) SDS], and the following processes were performed according to step 10–16 of R-ChIP protocol. Analysis of the hybridization efficiency was performed according to step 18 of the R-ChIP protocol.

**Analysis of decross-linked protein and chromatin during heat and hybridization.** Different hybridization buffers were used, including MS buffer, and LS buffer, respectively, supplemented with 10% (v/v) formamide, and 0, 0.5%, or 1% (w/v) SDS. A 100 μL aliquot was taken as sample 1. The remaining cross-linked DNA was treated at 90 °C for 2 min to denature the chromatin, incubated at 37 °C for 4 h for hybridization, and then a 100 μL aliquot was taken as sample 2. Samples 1 and 2 were extracted separately with an equal volume of Tris-phenol and chloroform (1:1, v/v), centrifuged at 12,000 × *g* for 10 min, and the supernatant was transferred into a new centrifuge tube. The supernatant was extracted with an equal volume of chloroform. The DNA in the supernatant was purified using a PCR purification kit (Qiagen), and used as a PCR template.

**The effects of different cross-linking methods on decross-linking during heat and hybridization processes.** To determine the effects of different concentrations of formaldehyde and Ni$^{2+}$ on decross-linking during heat treatment and hybridization, the chromatin were cross-linked using 1%, 3% (w/v) formaldehyde, or 3% (w/v) formaldehyde combined with 20 mg L$^{-1}$ NiSO$_4$, respectively. These cross-linked chromatins were incubated in LS buffer [containing 10% (v/v) formamide and 0.5% (w/v) SDS] at 90 °C for 2 min, and then incubated at 37 °C for 4 h. After incubation, the hybridization buffer was extracted following the method shown in last section and the extracted DNA were used as a PCR template.

**The effects of heat treatment duration on decross-linking.** The effects of heat treatment on decross-linking were analyzed. The chromatin was cross-linked using 3% (w/v) formaldehyde, and was divided with two equal portions. One portion was transferred to a 96-well cell plate and treated with ultraviolet ray at 254 nm with 7000 μJ cm$^{-2}$ for 10 min, and then used for hybridization (sample). At the same time, the other portion, without ultraviolet ray treatment, was used as control. Both sample and control were treated at 90 °C for 0, 3, and 9 min and 3 h in LS buffer [containing 10% (v/v) formamide and 0.5% (w/v) SDS]. After incubation, the hybridization buffer was extracted following the method shown above and the extracted DNA were used as a PCR template.

**Determination of the most suitable conditions for denaturation of cross-linked chromatin.** To determine the most suitable time for the denaturation of cross-linked chromatin at 90 °C, the chromatin was cross-linked with 3% (w/v) formaldehyde and 20 mg L$^{-1}$ NiSO$_4$, and treated with ultraviolet radiation for 10 min. After sonication, the cross-linked chromatin was added with an equal volume of SYBR Green Real-time PCR Master Mix (Takara) and incubated at room temperature for 10 min, incubated at 90 °C, and then florescence was detected at 5 s intervals in a real-time PCR machine qTOWER2.0 (Analytic, Jena, Germany). Five biological replicates were performed.

**The effect of hybridization duration on decross-linking of chromatin and proteins.** To determine the effect of hybridization duration on decross-linking, the formaldehyde-cross-linked chromatin were divided into two equal portions; one portion was transferred to a 96-well cell plate and treated with ultraviolet radiation at 254 nm with 7000 μJ cm$^{-2}$ for 10 min, and then used for heat and hybridization testing (sample). The other portion, without ultraviolet radiation treatment, was used as the control. Both sample and control were incubated at 90 °C for 2 min, and then at 37 °C in LS buffer [10% (v/v) formamide, 1 mM PMSF, proteinase inhibitors 1 μg mL$^{-1}$, supplement with 0.5% (w/v) SDS]. Aliquots of the hybridization products were taken after 0, 4, 8, and 12 h, and were extracted following the method shown above and the extracted DNA were used as a PCR template.

**The procedures for Reverse Chromatin Immunoprecipitation (R-ChIP).** About 10 g of plant material was used for the R-ChIP study, and divided into 4 equal portions. Each portion was processed by the following procedures, and the products were pooled together after the elution step. The procedures of R-ChIP were as follows: **Cross-linking**: (1) Each 2.5 g of materials were incubated in 30 mL of buffer A [10 mM Tris pH 8, 3% (w/v) formaldehyde, 20 mg L$^{-1}$ NiSO$_4$, 1 mM PMSF, 0.01% Silwet (as a wetting agent)] under a vacuum. After incubation for 20 min, 2 mL of 2 M Glycine was added, and incubated for 5 min to stop the cross-

linking reaction. **Nucleus purification**: (2) After cross-linking of chromatin, the samples were ground into a fine powder under liquid nitrogen, and then added with 30 mL of buffer B [10 mM Tris-HCl (pH 8.0), 0.4 M sucrose, 10 mM MgCl₂, 5 mM 2-mercaptoethanol, 1 mM PMSF, proteinase cocktail 1 µg mL⁻¹], shaken in a rotary shaker at 120 rpm for 5 min, and incubated on ice for 20 min. (3) The solution was filtered through two layers of Miracloth (Calbiochem, San Diego, CA, USA), and the filtered solution was centrifuged at $3000 \times g$ for 15 min at 4 °C. (4) Buffer C [10 mM Tris-HCl (pH 8.0), 0.25 M sucrose, 10 mM MgCl₂, 1% Triton X-100, 5 mM 2-mercaptoethanol, 1 mM PMSF, proteinase cocktail 1 µg mL⁻¹] was added to the precipitate and centrifuged at $14,000 \times g$ for 10 min at 4 °C. This procedure was repeated once. (5) The pellet was resuspended in 300 µL of buffer C (nuclei suspension). In a new centrifuge tube, 300 µL of buffer D [10 mM Tris/HCl (pH 8.0), 1.7 M sucrose, 0.15% (v/v) Triton X-100, 2 mM MgCl₂, 5 mM 2-mercaptoethanol, 1 mM PMSF, proteinase inhibitors 1 µg mL⁻¹] was added, and this layer was overlaid with the nuclei suspension and centrifuged for 30 min at $14,000 \times g$ at 4 °C. **Ultraviolet treatment**: (6) After centrifugation, the pellet was resuspended in 350 µL of lysis buffer [50 mM Tris-HCl (pH 8.0), 1 % (w/v) SDS, proteinase inhibitor cocktail 1 µg mL⁻¹ each] (the chromatin solution), and then transferred to a 96-well cell culture plate. The chromatin solution was treated with ultraviolet radiation at 254 nm with 7000 µJ cm⁻² for 10 min on the ice. After ultraviolet treatment, the chromatin solution was transferred into a new centrifuge tube. **Sonication**: (7) To shear the chromatin into DNA fragments, the chromatin solution was sonicated for 10 s, stopped for 20 s, repeated for 30 min at 90 W power setting. (8) The chromatin solution was centrifuged at $14,000 \times g$ for 10 min at 4 °C, and the supernatant was transferred to a new centrifuge tube. A 20 µL aliquot was taken as the input control (sample 1). **Precleaning of the chromatin**: (9) Magnetic beads (35 µL) were added to a clean centrifuge tube. The magnetic beads were harvested using the magnetic frame, and 350 µL of hybridization buffer was added to resuspend the magnetic beads, which were then harvested using the magnetic frame. The clean beads were added into the chromatin solution and incubated at 37 °C for 30 min, and then the beads were harvested using the magnetic frame. The precleaning process was repeated once. **Chromatin hybridization**: (10) An equal volume of 2× LS hybridization buffer [30 mM Tris-HCl (pH 7.4), 20% (v/v) formamide, 40 mM KCl, 20 mM (NH₄)₂SO₄, 6 mM MgCl₂, proteinase inhibitor cocktail 1 µg mL⁻¹] was added to the remaining sonicated chromatin solution, and then 0.6 µM of the mixed probes (20 nt in length, labeled with biotin at their 5′ termini) were added. (11) The above solution was mixed well and incubated at 90 °C for 2 min to denature the DNA, and then the tube was placed in a mixture of ice and water immediately. The solution was then incubated at 37 °C with 10 rpm rotation for 4 h for hybridization (hybridization product). **Isolation of hybridized DNA using magnetic beads**: (12) Magnetic beads (Dynabeads MyOne Streptavidin C1, Carlsbad, CA, USA) at 1 µL per pmol of probe were placed in a clean centrifuge tube. The beads were harvested using the magnetic frame to remove the supernatant, and washed once with 1000 µL of hybridization buffer. (13) The hybridization product was added to the magnetic beads, mixed well, and incubated at 37 °C for 2 h. (14) After incubation, the beads were harvested using the magnetic frame. **Bead washing**: (15) The magnetic beads were washed sequentially with the following buffers: low salt wash buffer [20 mM Tris pH 7.4, 0.1% (w/v) SDS, 150 mM NaCl, proteinase inhibitors 1 µg mL⁻¹] (washing once), high salt wash buffer [20 mM Tris (pH 7.4), 0.1% (w/v) SDS, 500 mM NaCl, proteinase inhibitors 1 µg mL⁻¹] (washing once), LiCl wash buffer [20 mM Tris (pH 7.4), 250 mM LiCl, proteinase inhibitors 1 µg mL⁻¹] (washing once), and Tris buffer [10 mM Tris (pH 7.4), proteinase inhibitors 1 µg mL⁻¹] (washing twice). Washing was accomplished by adding 1 mL of washing buffer to the magnetic beads, and placing the tubes on the magnetic stand for 1 min, and then removing the supernatant. **Elution**: (16) 300 µL of biotin elution buffer [12.5 mM biotin, 7.5 mM HEPES (pH 7.5), 75 mM NaCl, 1.5 mM EDTA, 0.15% (w/v) SDS, 0.075% (w/v) Sarkosyl, and 0.02% (w/v) Na-Deoxycholate] was added to the beads, which were incubated at room temperature for 30 min with rotation at 10 rpm, after which the beads were harvested using the magnetic frame. The supernatant was collected. Two hundred microliter of biotin elution buffer was added to the beads, which were incubated at 65 °C for 20 min, and centrifuged at $10,000 \times g$ for 1 min, and the supernatant was collected. The two supernatants were mixed together as 500 µL in total. Sixty microliters of the supernatant was taken as sample 2 for analysis. **Concentration of proteins**: (17) The products were concentrated using a protein filter (100 kd cutoff) to 30 µL, which was used for mass spectrometry analysis. **Analysis of the hybridization efficiency**: (18) Samples 1 and 2 were added with proteinase K, 0.2 M NaCl, 0.5% (w/v) SDS, and incubated at 55 °C for 3 h for decross-linking. After incubation, the chromatin DNA was purified using a PCR purification kit (Qiagen), and used as a PCR template to determine the hybridization efficiency assay. The gene promoter for *AtActin3* (AT3G53750) was used as the internal reference. The truncated promoter of *AtCAT3* was PCR amplified using primers, the DNA from samples 1 and 2 as templates.

**Mass spectrometry**. Mass spectrometry assay were carried out using an Orbitrap Mass Spectrometry (Q-Exactive Thermo Fisher, Waltham, MA, USA). Protein database searching was performed on Mascot (Matrix Science, v. 2.2) to obtain the qualitative identification information for the target protein and peptide molecules. The sequences of *Arabidopsis* proteins were retrieved from the uniprot_A._thaliana_89174_20200217 fasta file.

**ChIP analysis**. *Arabidopsis* plants were transiently transformed for expression of the TF and flag fusion genes, which were used for the ChIP study. The ChIP procedure was conducted following the method of Haring et al.[20]. Briefly, the chromatin and proteins were cross-linked with 1% (w/v) formaldehyde. After isolation of the cross-linked products, the chromatin was sonicated to shear it into 0.2–0.7 kb fragments. An anti-Flag antibody (Abmart, Berkeley Heights, NJ, USA) was used to immunoprecipitate the target chromatin (ChIP+), and chromatin without antibody immunoprecipitation served as a negative control (ChIP−). After reversal of cross-linking, the immunoprecipitated DNA was purified using a QIAquick spin column (Qiagen). The sites of *AtCAT3* promoter for ChIP-qPCR are provided in Supplementary Data 4. All qPCR primers are provided in Supplementary Data 8.

**Electrophoretic mobility shift assay (EMSA)**. The CDS of potential upstream regulators of *AtCAT3*, including *bZIP1664*, *TEM1*, *bHLH106*, *BTF3*, and *HAT1*, were cloned into the pMAL-c5X vector (NEB, Ipswich, MA, USA) to generate the recombinant vector expressing the proteins fused with the maltose-binding protein (MBP). The constructs were introduced separately into *Escherichia coli* strain ER2523 and were induced by isopropyl β–D-1-thiogalactopyranoside (IPTG) at 37 °C for 4 h. The fusion proteins were visualized using SDS-PAGE, and purified according to the instruction manual of the pMAL™ Protein Fusion & Purification System (NEB). The truncated promoters of *AtCAT3* that had been found to bind the target TFs according to the ChIP results were labeled with biotin. The same DNA sequence without biotin labeling was used as the competitor, and the promoter of *AtActin3* was used as the mutant probes. The interaction between the probe and the protein was performed using a Chemiluminescent EMSA kit (Beyotime, Jiangsu, China). The primers used for the construction of the recombinant pMAL-c5X vectors and the primers used for amplification of the DNA probes for EMSA are shown in Supplementary Data 6 and 7, respectively.

**Real-time PCR analysis**. The quantitative PCR experiments were performed on qTOWER2.0 (Analytic Jena, Germany). The real-time PCR reaction system contained 10 µL of SYBR Green Real-time PCR Master Mix (Takara), forward and reverse primers (0.5 µmol L⁻¹ each), and 2 µL of cDNA or DNA as the template, in a total volume of 20 µL. PCR was performed as the follows: 94 °C for 30 s; followed by 50 cycles at 94 °C 10 s, 59 °C 20 s, and 72 °C 30 s. The gene or promoter of *AtActin3* was used as the internal control. The sequences of primers are provided in Supplementary Data 8. Three independent biological replications were performed, and the relative expression levels were calculated according to $2^{-\Delta\Delta Ct}$ method[21].

**Statistics and reproducibility**. Data were compared using Student's *t*-test and two-way analysis of variance. Differences were considered to be significant if $P < 0.05$. Statistical analyses were carried out using SPSS 21.0 (SPSSInc, Chicago, IL, USA) software. One sample contains at least 500 individual *Arabidopsis* plants, and three biological replicates of R-chip were performed. The biological replicates are parallel measurements of biologically distinct samples.

**Reporting summary**. Further information on research design is available in the Nature Research Reporting Summary linked to this article.

## Data availability

The mass spectrometry proteomics data have been deposited to the ProteomeXchange Consortium via the PRIDE partner repository with the dataset identifier PXD020755 (wild-type samples) and PXD020772 (transiently transformed samples). Project Webpage: http://www.ebi.ac.uk/pride/archive/projects/PXD020755 and http://www.ebi.ac.uk/pride/archive/projects/PXD020772. FTP Download: http://ftp.pride.ebi.ac.uk/pride/data/archive/2020/11/PXD020755 and http://ftp.pride.ebi.ac.uk/pride/data/archive/2020/11/PXD020772. All source data underlying the graphs and charts in the main figures are provided as Supplementary Data 5.

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

## Acknowledgements

This work was supported by the High Level Talent Introduction Project of Xinjiang Uygur Autonomous Region, the National Natural Science Foundation of China (No. 31770704) and National Key Research and Development Project of China (2016YFC0501505).

## Author contributions

Y.W. and D.Z. planned and designed the research. X.W., J.W., and Y.D. performed experiments, X.J. and Z.T. analyzed data. Y.W. and X.W. wrote the manuscript.

## Competing interests

The authors declare no competing interests.
