## [Peer Review File · Communications Biology]

Reviewers' comments:

Reviewer #1 (Remarks to the Author):

The authors present a method to identify nuclear proteins contained in chromatin fragments of interest in plant (*Arabidopsis thaliana*) tissue. A similar method (Dejardin & Kingston, 2009) has been reported previously as acknowledged by the authors in the introduction and the discussion (see Comparison of PICh [Proteomics of Isolated Chromatin Segments] and R-ChIP). Chromatin proteins are immobilised by formaldehyde- and UV-assisted DNA crosslinking. Subsequently, crosslinked chromatin is randomly fragmented by sonication. Then, chromatin fragments of interest are selected by hybridisation using biotin-tagged oligonucleotide probes and subjected to mass-spectrometry (MS) for protein identification. The authors claim that their method can be applied to any chromatin segment including single-copy loci presumably because of improved conditions that reduce detachment of chromatin proteins through mechanical force or heat. This would be a step forward from the PICh method (Dejardin & Kingston, 2009) that requires hundreds of litres of cell culture to identify proteins associated with telomeric (repetitive) DNA sequences. I can clearly see some experimental benefit from testing different conditions to reduce protein loss. However, I am not sure whether the authors provide enough robust evidence that their method works (see Figure 7C).

With regard to statistics, all datapoints should be incorporated into bargraphs as dotplots (e.g. Figure 1B). Also please indicate the number of biological replicates for each graph and the results of each statistical test in the figure legend.

Figure 7C

- What is the negative control in Figure 7C?
- How was the enrichment calculated? Add reference to the figure legend.
- Were the TFs overexpressed for ChIP-qPCR?
- I would prefer the ChIP results were presented as a percentage of input rather than using another genome locus as reference!
- Site 1 and 7 are overlapping. How is it possible that these sites yield different ChIP enrichments given that the fold change reference is the same? This does not seem credible. The resolution of ChIP-qPCR depends on chromatin shearing, which produces a broad range of DNA fragments from a few hundreds to thousands of base pairs. Thus, ChIP-qPCR does not discriminate DNA binding and non-binding events across a few hundred base pairs.
- How were the genomic sites selected for ChIP-qPCR? E.g. for ZFP2 sites 1, 7 and 8 were chosen, for TEM1 only 7.

In general, the enrichment values are rather low and sometimes conflicting each other (see above). I am not convinced that they show genuine TF binding.

Legend of Figure 3: deltaCt is simply the difference to a reference Ct (not variation)

Legend of Figure 6:

- Line 885: ratios 1 and 2 are deltaCts, thus call them deltaCt1 and deltaCt2
- Line 891/2: technical replicates (not technological replications)

Raw files attached to Project PXD014911 submitted to ProteinXchange were accessible with their credentials.

Reviewer #2 (Remarks to the Author):

In this manuscript, the authors described a DNA-centered method, which could successfully isolate regulatory proteins from a single locus of a gene in the transiently transformed plant containing

truncated CAT promoter. The method is novel in the sense of using a transiently transformed promoter, which increases the possibility to capture associated-proteins from a single locus. However, it is difficult to see that the method and the finding are breakthrough. Firstly, the main technique used in R-ChIP is similar to that in PICh, although there are some minor differences. In addition, the plant version of PICh has been reported in barley (Zeng et al., 2016, Journal of Proteome Research, 15:1875-1882). Secondly, most of the results provided in the manuscript are showing the optimization of the R-ChIP. Evidence for the regulation of AtCAT is largely missed making this manuscript mainly focus on technique improvement. Lastly, the analyses for R-ChIP identified DNA-associated proteins are descriptive and the authors need to provide more information and experimental evidence.

Major comments

1. The manuscript started with optimization of the R-ChIP procedure by modifying each critical factor and introduced R-ChIP after, which makes the manuscript confusion and hard to follow. The main text lacks key information of the probe features (location, length). I would suggest that the authors should reorganize the manuscript.
2. The authors concluded good repeatability among three technique replications of R-ChIP by visualizing protein bands on SDS-gel. However, it seems that only one sample from each material was used for MS analysis. It would be more accurate and convincing if biological replicates are used for MS.
3. The authors measured the change of the amount of protein-associated DNA during treatment to indicate the effectiveness of possible recovery of DNA-associated proteins. However, the change of the amount of DNA may not reflect the diversity of the proteins. The authors should include a few MS results from other combinations of R-ChIP conditions (buffer, SDS, temperature).
4. ChIP-qPCR analyses for R-ChIP identified TFs were performed using TT plants, which contains transiently transformed AtCAT promoter. I would suggest to examine the binding of TFs to the endogenous AtCAT promoter.

Minor comments

-Figure 1

The authors indicated successfully transient transformation by using empty pCAMBIA1301, while the efficiency of transient transformation was shown using qRT-PCR for detecting pBI121-ProCAT:AtCAT3-GUS. It would be nice to see the GUS-staining result from the latter transformation.

Please provide the internal control for the evaluation of relative expression in the figure legend or Method section.

-Figure 3

It's not clear what gene/DNA fragment was used for the qPCR evaluation.

-Figure 4a

This figure needs to be reorganized. The authors did not mention the result of "+UVB" from Figure 4a in the text (line 163-182).

-Line 197-198: the conclusion could not be drawn from the results.

-Line 212: rewrite the sentence "add a set of single strain DNA probes was added to ...".

-Figure 5

This figure looks unclear to me. It seems that there are two possible conformation of denatured-chromatin. Please provide more detailed legend to explain the procedure or make the figure more understandable.

-Figure 6a, line 887-line889, please specify which internal reference was used for WT and which for TT, since there are two internal references mentioned together in the legend.

-Line 236, "and more proteins were isolated from TT than from WT plants". Please explain why more proteins were isolated from TT than WT by the SDS-PAGE analysis.

-The order of Figure 7 should be reorganized, since Fig. 7A appears in the Method.

Response

Reviewer #1 (Remarks to the Author):

The authors present a method to identify nuclear proteins contained in chromatin fragments of interest in plant (*Arabidopsis thaliana*) tissue. A similar method (Dejardin & Kingston, 2009) has been reported previously as acknowledged by the authors in the introduction and the discussion (see Comparison of PICh [Proteomics of Isolated Chromatin Segments] and R-ChIP). Chromatin proteins are immobilised by formaldehyde- and UV-assisted DNA crosslinking. Subsequently, crosslinked chromatin is randomly fragmented by sonication. Then, chromatin fragments of interest are selected by hybridisation using biotin-tagged oligonucleotide probes and subjected to mass-spectrometry (MS) for protein identification. The authors claim that their method can be applied to any chromatin segment including single-copy loci presumably because of improved conditions that reduce detachment of chromatin proteins through mechanical force or heat. This would be a step forward from the PICh method (Dejardin & Kingston, 2009) that requires hundreds of litres of cell culture to identify proteins associated with telomeric (repetitive) DNA sequences. I can clearly see some experimental benefit from testing different conditions to reduce protein loss.

1. With regard to statistics, all datapoints should be incorporated into bargraphs as dotplots (e.g. Figure 1B). Also please indicate the number of biological replicates for each graph and the results of each statistical test in the figure legend.

Response: We have incorporated datapoints into bargraphs as dotplots in Figure 2 and 3 according to your suggestion. However, Figure 1B (Fig. 5B in revision) has not been changed into bargraphs for the reason that it shows the expression of transformed (TT) plants relative to WT plants, and is not suitable to be shown as dotplots. The number of biological replicates and the statistical test have been included in the corresponding figure legend.

2. However, I am not sure whether the authors provide enough robust evidence that their method works (see Figure 7C).

- What is the negative control in Figure 7C?

Response: The negative control showed in original version was a fragment of the promoter of gene *AtActin3* (*AtActin3* and *AtCAT3* are located in different chromosomes). We have performed the ChIP experiment again, and using % input to determine the enrichment, we used ChIP- as the negative controls, and the original negative controls were deleted from the revision.

- How was the enrichment calculated? Add reference to the figure legend.

Response: The ChIP enrichment was calculated using an internal control method according to Haring et al. (2007). According to your suggestion, ChIP enrichment was shown as % input. Therefore, the reference for using an internal control method was not provided in the revision.

Ref: Haring M, Offermann S, Danker T, Horst I, Peterhansel C, Stam M. Chromatin

immunoprecipitation: optimization, quantitative analysis and data normalization. *Plant Methods*. 2007;3:11.

- Were the TFs overexpressed for ChIP-qPCR?

Response: Yes, these TFs were transiently transformed into *Arabidopsis* plants for overexpression, and the transiently transformed (TT) plants were used for ChIP-qPCR.

- I would prefer the ChIP results were presented as a percentage of input rather than using another genome locus as reference!

Response: The ChIP results were shown as % input according to the suggestion.

- Site 1 and 7 are overlapping. How is it possible that these sites yield different ChIP enrichments given that the fold change reference is the same? This does not seem credible. The resolution of ChIP-qPCR depends on chromatin shearing, which produces a broad range of DNA fragments from a few hundreds to thousands of base pairs. Thus, ChIP-qPCR does not discriminate DNA binding and non-binding events across a few hundred base pairs.

Response: Thank you for your comments! Yes, site 1 and 7 are overlapping. However, compared with site 1, the site 7 contains more different cis-acting elements, and these cis-acting elements enable that some TFs may bind to site 7 rather than binding to site 1. We agree that ChIP-qPCR does not discriminate DNA binding and non-binding events across a few hundred base pairs. However, considering that site 7 contains more cis-acting elements than site 1, the abundances of DNA binding and non-binding fragments in site 7 should be different with site 1 after sonication. This phenomenon is just like the peaks identified by ChIP-seq, and some sequences may be in the peaks both at site 7 and site 1, but the peak values in site 1 and 7 should be different. In addition, we have performed the ChIP-qPCR again, and the results also showed that the enrichment of site 1 and 7 were difference. Thank you!

- How were the genomic sites selected for ChIP-qPCR? E.g. for ZFP2 sites 1, 7 and 8 were chosen, for TEM1 only 7.

Response: All the sites have been analyzed using ChIP-qPCR, but only the sites that were significantly enriched have been shown in the figure. The reason that not all the sites can be significantly enriched are that some sites for ChIP-qPCR are far away from the binding site of studied TFs, and could not be enriched by ChIP.

-In general, the enrichment values are rather low and sometimes conflicting each other (see above). I am not convinced that they show genuine TF binding.

Response: These data were shown as $\Delta\Delta Ct$, and not the real enrichment fold. For more reliability, we have performed ChIP-qPCR again, and these data were analyzed using % input according to the suggestion, and the enrichment were from 6.1 to 111.4.

3. Legend of Figure 3: deltaCt is simply the difference to a reference Ct (not variation).

Response: Yes, deltaCt is the difference to a reference Ct, which was used for reflecting the amount of decross-linking of chromatin after hybridization. We have

changed “Ct value variation” with “Ct value difference” in the revision.

4. Legend of Figure 6:

- Line 885: ratios 1 and 2 are deltaCts, thus call them deltaCt1 and deltaCt2.

Response: Ratios 1 and 2 has been changed with deltaCt1 and deltaCt2 according to your suggestion (line 792).

- Line 891/2: technical replicates (not technological replications).

Response: We have corrected this mistake according to the suggestion (line798/800).

5. Raw files attached to Project PXD014911 submitted to ProteinXchange were accessible with their credentials.

Response: According to the suggestion, the R-ChIP has been performed again, and the raw data have been submitted to ProteomeXchange, and Project accession numbers were PXD020772 and PXD020755.

Reviewer #2 (Remarks to the Author):

In this manuscript, the authors described a DNA-centered method, which could successfully isolate regulatory proteins from a single locus of a gene in the transiently transformed plant containing truncated CAT promoter. The method is novel in the sense of using a transiently transformed promoter, which increases the possibility to capture associated-proteins from a single locus. However, it is difficult to see that the method and the finding are breakthrough. Firstly, the main technique used in R-ChIP is similar to that in PICh, although there are some minor differences. In addition, the plant version of PICh has been reported in barley (Zeng et al., 2016, Journal of Proteome Research, 15:1875-1882). Secondly, most of the results provided in the manuscript are showing the optimization of the R-ChIP. Evidence for the regulation of AtCAT is largely missed making this manuscript mainly focus on technique improvement. Lastly, the analyses for R-ChIP identified DNA-associated proteins are descriptive and the authors need to provide more information and experimental evidence.

Major comments:

1. The manuscript started with optimization of the R-ChIP procedure by modifying each critical factor and introduced R-ChIP after, which makes the manuscript confusion and hard to follow. The main text lacks key information of the probe features (location, length). I would suggest that the authors should reorganize the manuscript.

Response: Yes, we agree that it is hard to follow, and the manuscript has been reorganized according to your suggestion. To reorganize the manuscript, we have moved the procedures of R-ChIP (Fig. 5 in original version) first (shown as Fig. 1 in revision), and moved the section of “Determination of the efficiency of transient transformation” after the section of “Building a procedure for reverse ChIP”.

The key information of the biotin probe features used for hybridization has been provided in supplementary Figure S1, and also been specified in the text (line 243-245). Thank you!

2. The authors concluded good repeatability among three technique replications of R-ChIP by visualizing protein bands on SDS-gel. However, it seems that only one sample from each material was used for MS analysis. It would be more accurate and convincing if biological replicates are used for MS.

Response: We have performed three biological repeats in the revision, and the results have been shown in the manuscript. Most TFs that had been identified in the original manuscript were also identified in this study, indicating that the results were reliable. The repetition rates between two biological repeats were 55.7-62.2% in WT samples and 37.4-45.3 % in TT samples. We have discussed the reason for low repetition rates between two biological repeats in the revision (line 382-399).

3. The authors measured the change of the amount of protein-associated DNA during treatment to indicate the effectiveness of possible recovery of DNA-associated proteins. However, the change of the amount of DNA may not reflect the diversity of the proteins. The authors should include a few MS results from other combinations of R-ChIP conditions (buffer, SDS, temperature).

Response: Yes, MS results can reflect the diversity of the proteins. However, there were some problems in MS protein analysis. Because the low abundance of TFs, the biological repeats of MS are not good. In addition, as our MS results shown, there were some contamination proteins (background) in R-ChIP, which will yield false results. These factors together will make the analysis of protein diversity to be difficult. Therefore, the MS results may not be reliable in determination of diversity of proteins. In addition, compared with MS analysis, protein-associated DNA can be quantitatively studied, which could tell the small difference between different treatments, enabling us to choose the best treatment; however, MS results could not be quantitatively studied as measurement of protein-associated DNA. This is the reason that MS results from other combinations of R-ChIP conditions were not studied and provided.

4. ChIP-qPCR analyses for R-ChIP identified TFs were performed using TT plants, which contains transiently transformed *AtCAT* promoter. I would suggest to examine the binding of TFs to the endogenous *AtCAT* promoter.

Response: ChIP-qPCR was performed with the plants that transiently transformed the aim TF (i.e. the TF that were identified to be the regulators of *AtCAT3*), and the promoter of *AtCAT3* was not transformed. Therefore, these ChIP-qPCR have examined the binding of TFs to the endogenous *AtCAT3* promoter, but not the transformed *AtCAT3* promoter.

Minor comments:

1. Figure 1:

-The authors indicated successfully transient transformation by using empty pCAMBIA1301, while the efficiency of transient transformation was shown using

qRT-PCR for detecting pBI121-ProCAT:AtCAT3-GUS. It would be nice to see the GUS-staining result from the latter transformation.

Response: PBI121 vector is not suitable for GUS staining when using transient transformation, because GUS staining in *Agrobacterium* cells harboring pBI121 can be detected highly. For the reason that the *GUS* gene in pBI121 do not have intron, which can express GUS activity in prokaryotic cells. At the same time, there were *Agrobacterium* cells contamination throughout the transiently transformed plants, which will be detected by GUS staining, leading to false result. Therefore, we studied the expression of *AtCAT3* using qPCR instead of using GUS staining.

-Please provide the internal control for the evaluation of relative expression in the figure legend or Method section.

Response: Internal control used in this figure was *AtActin3*, we have explained it in the revision, and its primer sequences have been shown in supplementary Table S6. We also provided more detail information in the figure legend (line 784-786).

2. Figure 3: It's not clear what gene/DNA fragment was used for the qPCR evaluation.

Response: We use the truncated promoter of *AtCAT3* for the qPCR evaluation. The location and length information of the site were shown in Figure 7L and supplementary Table S3, and the primer sequences were shown as supplementary Table S6 (Q-CAT3p-1-F/R). We have provided detail information in the figure legend (line746-747) and materials and methods (line674-675).

3. Figure 4a:

-This figure needs to be reorganized. The authors did not mention the result of "+UVB" from Figure 4a in the text (line 163-182).

Response: We have reorganized the text, and the result of "+UVB" from Figure 4A were explained at the section of "Heat and hybridization induces decross-linking that can be alleviated using ultraviolet radiation crosslinking" (line 187-198). Thank you!

-Line 197-198: the conclusion could not be drawn from the results.

Response: Yes, the conclusion could not be drawn from the results in the original revision. We have added the results of UVB treatment on decross-linking of DNA and protein, enabling this conclusion could be drawn from the results (line 193-198).

5. Line 212: rewrite the sentence "add a set of single strain DNA probes was added to ...".

Response: This sentence has been modified as "a set of single strain DNA probes were added to ..." (line212-213).

6. Figure 5: This figure looks unclear to me. It seems that there are two possible conformation of denatured-chromatin. Please provide more detailed legend to explain the procedure or make the figure more understandable.

Response: We have modified this figure for more clearly, and conformation of denatured-chromatin has been unified (Fig. 1 in the revision); in addition, we have

provided more detailed legend to explain the procedure according to the suggestion (line704-712).

7. Figure 6a, line 887-line889, please specify which internal reference was used for WT and which for TT, since there are two internal references mentioned together in the legend.

Response: WT and TT used the same internal reference, which is the promoter of *AtActin3*, and we have modified the figure legend to make it more clearly (line 796-797).

7. Line 236, “and more proteins were isolated from TT than from WT plants”. Please explain why more proteins were isolated from TT than WT by the SDS-PAGE analysis.

Response: Yes, SDS-PAGE could not show protein abundance clearly, and the conclusion “more proteins were isolated from TT than from WT plants”, and we have deleted the sentence in the version.

8. The order of Figure 7 should be reorganized, since Fig. 7A appears in the Method.

Response: We have reorganized the order of Figure7A, which was shown as Figure 7L. Thank you!

REVIEWERS' COMMENTS:

Reviewer #1 (Remarks to the Author):

The authors present a method, R-ChIP to identify nuclear proteins on regulatory DNA in transiently transformed plant tissue. A similar method, PiCh (Dejardin & Kingston, 2009), has been reported previously as appropriately acknowledged by the authors. In R-ChIP, chromatin proteins are immobilised by formaldehyde- and UV-assisted DNA crosslinking. Subsequently, crosslinked chromatin is randomly fragmented by sonication. Eventually, chromatin fragments of interest are selected by hybridisation using biotin-tagged oligonucleotide probes and subjected to mass-spectrometry (MS) for protein identification. The authors claim that their method can be applied to any chromatin segment including single-copy loci because of improved conditions that reduce the detachment of proteins from chromatin. I consider this a step forward from the PiCh method (Dejardin & Kingston, 2009) that requires hundreds of litres of cell culture to identify proteins associated with telomeric (repetitive) DNA sequences.

Reviewer #2 (Remarks to the Author):

In the revised manuscript, the authors have addressed most of my previous concerns. I appreciate that the authors supplemented additional experiments and data. I only have following minor comments:

1. "cis" should be italic in the entire manuscript.
2. Line 75-80. This study in Arabidopsis has obvious advantages over previously developed gene-centered methods, such as PiCh in HeLa cell and in plant barley. However, this is not the first gene-centered method developed in plant system. The authors should acknowledge the effort in plant as well.
3. Figure 1 The diagram in the step "Heat for denaturing chromatin" is misleading, since the DNA-bound proteins are completely lost from chromatin where the probe binds. If this is the case, only proteins from undenatured chromatin can be isolated.
4. Figure 7 I expect to see similar enrichment of either TFs between ChIP+ and ChIP- in or around CAT3 promoter regions where no tested TFs bind. For example, the region between site 5 and site 8. Please provide information.

Response

Reviewer #2 (Remarks to the Author):

1. "cis" should be italic in the entire manuscript.

Response: The "cis" in the manuscript had all been italic according to your suggestion. (line41, 59, 406)

2. Line 75-80. This study in *Arabidopsis* has obvious advantages over previously developed gene-centered methods, such as PICh in Hela cell and in plant barley. However, this is not the first gene-centered method developed in plant system. The authors should acknowledge the effort in plant as well.

Response: Thank you for your comments! Yes, PICh had been performed in barley. We had added the related article and acknowledged this effort in the version. (line72-74)

3. Figure 1 The diagram in the step "Heat for denaturing chromatin" is misleading, since the DNA-bound proteins are completely lost from chromatin where the probe binds. If this is the case, only proteins from undenatured chromatin can be isolated.

Response: As DNA and protein had been crosslinked in this experiment, DNA-bound proteins can not be lost from chromatin where the probe binds. To avoid the misleading, the step "Heat for denaturing chromatin" had been changed into "Heat for denaturing DNA" based on the suggestion.

4. Figure 7 I expect to see similar enrichment of either TFs between ChIP+ and ChIP- in or around CAT3 promoter regions where no tested TFs bind. For example, the region between site 5 and site 8. Please provide information.

Response: According to your suggestion, all the enrichment fold of five TFs in or around CAT3 promoter regions had been shown in Supplementary Data 3.